# Clinical Characteristics in Early Childhood Associated with a Nevus-Prone Phenotype in Adults from Tropical Australia: Two Decades of Follow-Up of the Townsville Preschool Cohort Study

**DOI:** 10.3390/ijerph17228680

**Published:** 2020-11-23

**Authors:** Ramez Barsoum, Simone L. Harrison

**Affiliations:** 1Skin Cancer Research Unit, College of Public Health, Medical and Veterinary Sciences, James Cook University, Townsville 4811, Australia; Ramez.Barsoum@health.qld.gov.au; 2College of Medicine and Dentistry, James Cook University, Townsville 4811, Australia; 3Princess Alexandra Hospital, Queensland Health, Woolloongabba, Brisbane 4102, Australia

**Keywords:** melanocytic nevi, melanoma, sunlight, ultraviolet radiation, prevention, longitudinal, cohort, sun-protection, screening, childhood

## Abstract

Having numerous melanocytic nevi increases melanoma risk. Few studies have enumerated nevi in children and re-examined them as adults. We aimed to determine if childhood nevus-counts predict nevus-prone adults, and further explore the relevance of host-factors and sun-exposure. Fifty-one Caucasian residents of Townsville (19.16° S, Queensland, Australia) had full-body nevus-counts aged 1–6 and 21–31 years-old. Sun-exposure was determined from questionnaires. Children in the upper-quartile of nevus-counts acquired nevi more rapidly than those in the bottom-quartile (13.3 versus 4.7 nevi/year; *p* < 0.0005). Children sunburnt before 7 years-old acquired more incident nevi by adulthood (238 versus 126, *p* = 0.003) particularly if sunburn was severe (321 versus 157.5, *p* = 0.003) or erythema occurred annually (380 versus 132, *p* = 0.008). Fair-skinned, freckled children with some nevi ≥ 3 mm, solar lentigines, or a family history of melanoma acquired more incident nevi than children without these attributes. Nevus-prone adults exhibit distinguishing features earlier in life (<7 years-old in Queensland) than has been shown previously. In addition to intervening with sun-protection counselling early enough to reduce risk, being able to reliably triage children into high- and low melanoma-risk groups may inform more efficacious and cost-effective targeted-screening in melanoma-prone populations. Further longitudinal research is needed to confirm that these attributes can reliably separate risk-groups.

## 1. Introduction

Cutaneous melanoma is the fourth most common cancer in Australia (excluding keratinocyte cancers), and the most commonly diagnosed cancer in the 15–39 year age group [1]. Queensland, Australia has the highest incidence of malignant melanoma globally, with 71/100,000 cases diagnosed annually from 2009–2013 [2]. Although 5-year survival rates for melanoma in Australia are 90% or better [1], 1429 melanoma-related deaths were recorded in 2018 [1].

There is a causative link between sun-exposure experienced before 10 years-old and melanoma [3]. Numerous case-control studies have demonstrated that the number of melanocytic nevi acquired by adulthood is a strong independent risk-factor for melanoma [4], with sun-exposure implicated in nevus development during childhood [5,6,7]. Consequently, Caucasian children raised in Queensland’s intense ultraviolet radiation (UV) environment develop nevi earlier and in higher numbers than those from regions with less UV [5,6,7,8,9,10]. With increased focus on children’s sun-safe behaviors [11], and randomized controlled trials demonstrating that a significant proportion of the nevi that develop during childhood can be prevented with rigorous sun-protection [12,13], it is timely to explore the link between childhood and adult nevus-counts, and its implications for primary prevention and targeted-melanoma screening.

Few prospective cohort studies have re-examined child participants for nevi as adults. One small study (*n* = 40) re-examined child-participants for nevi after 20 years, but was not population-based as all participants were the offspring of malignant melanoma patients [14]. Consequently, little is known about the relationship between nevus-counts during early childhood and adulthood in the general population. The present study sought to address this knowledge gap by performing serial nevus-counts spanning 20+ years in a representative population-based pediatric cohort from tropical Australia to determine whether childhood nevus-counts could predict nevus-prone adults in the general population. Understanding this relationship may reveal clinically-distinguishable features that can be used to identify individuals at high risk of melanoma early enough in life for preventive strategies to be implemented to reduce risk.

## 2. Materials and Methods

Townsville (latitude 19.16° S), the largest city in regional Queensland (population 229,031) [15] is situated in the dry tropics. It experiences hot, humid summers, dry winters [16], and a high to extreme UV-index year-round [17]. We assessed melanocytic nevi, sun-exposure and phenotype in 2016 for a cohort of 21–31-year-old Caucasian participants (with at least 3 grandparents of European descent) who were still residing in Townsville, Australia more than 20-years after their nevi were first counted, to determine whether childhood nevus-counts predict nevus-prone adults. Guidelines for Strengthening the Reporting of Observational Studies in Epidemiology (STROBE) were used to describe the study protocol and sample [18]. 

The original cohort included 1–6-year-olds (born 1985–1994) recruited from the labor-ward records of Townsville’s maternity hospitals [5,6,8], whose parents provided written informed consent (87.6% response) for them to be examined for melanocytic nevi of all sizes (5 categories: <2 mm; 2 mm; 3 mm; 4 mm; ≥5 mm in diameter) at 30 body-sites (excluding the genitals, buttock, and scalp) [5,8] by an examiner (SLH) experienced in the recognition of pigmented lesions [5,6,8,19]. Parents completed a questionnaire detailing their child’s ethnicity, demographic characteristics, sun-sensitivity, and sun-exposure history. Nevus-counts were repeated 12 months later and parents completed another sun-exposure questionnaire on their child’s behalf [6]. 

Contact details were sought for 230 (approximately 50%) of the 484 Caucasian participants whose nevi were counted at baseline and 12 months later to determine which of them were still residing locally and to invite them to be re-examined for melanocytic nevi by RB in Townsville between January and June 2016. A variety of methods were used including: perusing social media; contacting parents by phone or mail to establish contact with the participant when baseline records included a valid telephone number or address; perusing residential telephone directories and the electoral roll for the Division of Herbert. A Local Television News story also aired in February 2016 encouraging past-participants to contact us. 

RB (currently a Dermatologist in training) conducted follow-up nevus-counts after being trained to recognize melanocytic nevi both macroscopically and dermoscopically by Dermatologists affiliated with the University of Queensland and Princess Alexandra Hospital. All lesions were examined with a Heine Delta 20T dermatoscope. To minimize inter-observer variability in nevus-counts, RB was also trained to follow the standard international protocol for the examination of melanocytic nevi [19] by SLH who examined the original cohort for nevi as children [5,8] with the assistance of a General Practitioner (Dr Beverly Raasch) with 40 years of experience in skin cancer medicine. The Director of Dermatology (Professor H.P. Soyer) at Princess Alexandra Hospital was also available to guide RB on the differential diagnosis of pigmented lesions through teledermatology and Dr Raasch was available locally to be consulted as necessary. 

At the follow-up examination conducted during adulthood, all melanocytic nevi were counted and recorded by size according to their widest diameter (3 categories: <2 mm, 2–4 mm and ≥5 mm) for 30 body-sites, excluding the scalp, buttocks, and genitals [19] as at baseline [5,8], except for some of the skin covered by the brassiere in women. Because melanocytic nevi ≥ 5 mm are clinically significant in relation to melanoma [4], all nevi ≥ 5 mm were photographed individually after an adhesive millimeter label with the participant’s unique identification number was placed in the field-of-view. Photographs of all nevi ≥ 5 mm were reviewed by Professor Soyer to minimize the risk of missed melanoma among participants, and due to his concurrent interest in these lesions in relation to training an artificial intelligence neural network. The posterior-trunk of each participant was also photographed. 

Melanocytic nevi < 2 mm were distinguished from freckles clinically based on their body-site, presence throughout the year even in the absence of solar stimulation [19,20] and dermoscopic features. Small nevi (<2 mm) were identified separately to facilitate direct comparison with studies enumerating nevi of all sizes [5,6,8,12] as well as older studies which only documented nevi ≥ 2 mm [9,10,13,14] or ≥ 3 mm in diameter [20]. Our decision to count nevi of all sizes is also justified in light of recent evidence about “micro-melanomas” with small diameters [21]. While nevi ≥2 mm are easier to distinguish from lentigo simplex and ephelides (freckles) than smaller nevi [20], deliberately excluding nevi < 2 mm from nevus counts may be misleading and cause problems interpreting data, as a considerable proportion of smaller, newly arisen nevi not counted at baseline would be (mis)classified as ‘incident nevi’ at subsequent examinations, when they have been present long enough (prevalent nevi) to grow to 2 mm and meet the size criterion for inclusion. This delay in identifying ‘incident nevi’ may influence the conclusions drawn, particularly regarding latency. The number of nevi < 2 mm that are intentionally excluded from nevus-counts in studies that adhere to the minimum 2 mm size criterion is not available to be added to earlier nevus counts, and can therefore under-estimate nevus frequency. In contrast, longitudinal studies that document nevi of all sizes, including those smaller than 2 mm, have the advantage of prospective determination of all individual pigmented lesions over time.

Skin reflectance of the sun-protected left-inner-upper-arm was measured using a reflectance spectrophotometer (Konica Minolta Optics Inc™ CM-2600d, Osaka, Japan) [22]. Constitutional skin color was categorized as fair, medium or olive and eye color was visually assessed (as brown, hazel or blue) using iris photographs from the baseline study [5,8]. Participants selected their natural hair color at age 18 years (black, dark-brown, light-brown, blond, red) from wig-samples used at baseline [5,8]. Height was measured using a wall-mounted stadiometer (Seca™, Hamburg, Germany) and weight in kilograms was measured using zero-calibrated 150 kg scales (Seca™, Hamburg, Germany). 

At baseline, the participants’ parents completed a comprehensive questionnaire describing their child’s demographic characteristics, ethnicity, sun-sensitivity, and sun-exposure history [5]. Interim sun-exposure was collected in a second questionnaire completed at their child’s 12-month nevus examination in 1992 [6]. The participants themselves also completed a self-administered questionnaire at the 20+ year follow-up to document familial and personal history of melanoma; history of epithelial skin cancer and excised nevi; and to seek additional information on their sun-exposure and sun-protection habits as a school student, during early adulthood and in their twenties (including recreational and occupational sun-exposure). 

The relationship between total nevus-count as a child and an adult (prevalence) was assessed using Pearson’s correlation coefficient (r) and paired t-tests, while Spearman’s rank correlation coefficient (Rho) was used to estimate the association between each participant’s rank (quartile of nevus prevalence within age group) at baseline and follow-up. The main outcome variable was incident nevus-count at follow-up (total nevi in 2016 minus total at baseline). Because the distribution of incident nevi was positively-skewed, medians and interquartile ranges (IQR) are provided unless otherwise stated. Bivariate relationships between the main outcome variable and phenotypic and sun-exposure variables were explored using Mann–Whitney (MW) and Kruskal–Wallis Tests (KW) to compare the distribution of nevi across two (MW) or multiple categories (KW) with a significance level of α = 0.05. The *p*-values for all univariate tests were ranked from smallest to largest and the Benjamini–Hochberg (BH) Procedure [23] was used to decrease the false positive rate (Type I error) due to multiple testing. We present unadjusted *p*-values in the tables, and use footnotes to indicate which results are statistically significant after adjustment for multiple testing with a false discovery rate of 10%. Statistical analysis was performed using SPSS™ statistical software version 23 for Windows (IBM SPSS, Inc., Chicago, IL, USA).

Approval was obtained from the Human Research Ethics Committee of James Cook University (H6190). Participation was voluntary. Participants received an information sheet explaining the study and signed a consent-form prior to examination. Data and images were de-identified (unique ID-number) to protect the identity of participants, and stored on a password-secured drive or in locked filing cabinets [24].

## 3. Results

### 3.1. Response

Of the 230 Caucasian participants whose details were sought and/or contact attempted, two were deceased and 97 moved away from Townsville (6 were overseas; 14 were interstate; 11 lived in Far North or rural Queensland; 45 lived in the greater Brisbane area, while relatives of 21 participants confirmed they had “left Townsville”). Twelve (9.2%) of the remaining 131 eligible participants declined to participate due to time-constraints or disinterest. Forty participants thought to reside locally were uncontactable within the required timeframe. This is not surprising given that some females in the original cohort will have changed surnames, making them more challenging to trace. Fifty-one (39% response) of the 131 eligible participants provided written informed consent to participate and were re-examined for nevi between Jan and June 2016.

### 3.2. Sample Characteristics

All 51 of these Townsville-born participants had least 3 grandparents of European origin and had been examined for melanocytic nevi at baseline when aged 1–6 years-old by SLH (Table 1). The sample of 51 adult Townsville residents that was followed-up was similar to the original Townsville Preschool Cohort [6] in terms of demographic, phenotypic and sun-exposure characteristics recorded at baseline (Appendix A).

The median age of eligible participants at baseline was 3.2 years [IQR, 2.3, 4.9] with a median duration of follow-up of 24 years [IQR, 23.8, 24.3] and median age at follow-up in 2016 of 27 years [IQR, 26, 29]. The majority of participants had light eyes, fair skin and dark hair at follow-up (Table 2). Most participants (82.4%) were working full-time or part-time, with 54.9% employed in predominantly indoor occupations. One participant (2%) had been diagnosed with melanoma in-situ (level 0) on the posterior-trunk five years earlier. A further 39.2% of participants had already had at least one nevus excised and 7.8% had been treated for keratinocyte cancer in the previous two years. The median number of incident nevi acquired between examinations was 185 [IQR, 107, 289] (Appendix A).

### 3.3. Predictors of Incident Nevus-Counts

The presence and extent of freckling (combined for face, forearms, and shoulders), and solar lentigines (across the shoulders) both as a child and as an adult were significant predictors of nevus-prone adults (Table 2). Participants who had at least one nevus ≥ 3 mm in diameter at baseline acquired more new nevi by adulthood than participants whose nevi were ≤2 mm at baseline. Fair skin color was also significantly associated with higher numbers of melanocytic nevi (all sizes and ≥2 mm in diameter) at follow-up (Table 2). Self-reported family history of melanoma also seemed to be associated with the acquisition of small nevi between examinations, while gender had no effect (Table 2).

The rate of development of new nevi between examinations was highest for participants who were in the upper-quartile of nevus-counts for age (prevalence) at baseline (Table 3). Accordingly, a strong, statistically significant correlation was evident between total nevus-count at baseline (i.e., nevus prevalence as a child) and at follow-up (i.e., nevus prevalence as an adult) (r = 0.704, *p* < 0.0005; Paired *t*-Test *t* = 10.194, *p* < 0.0005).

Participants who experienced a sunburn before age 7 years acquired significantly more new nevi of all sizes between examinations than those who were not sunburnt as a “preschool” child (Table 4). The influence of acute sun-exposure on nevus development was most evident for participants who experienced erythema annually during the “preschool” years, and in those where at least one sunburn during early childhood caused peeling or blistering. There was little evidence of an association with habitual sun-exposure during or since leaving school. Only moderate (2–4 h/day) recent weekday sun-exposure produced a significant result (Table 4).

## 4. Discussion

To our knowledge, this is the first population-based study to re-examine a cohort for melanocytic nevi more than two decades after examining them as young children. Our results demonstrate that young children with high nevus-counts tend to become nevus-prone adults, with nevus-counts during early childhood being a good predictor of nevus-count in the third decade, when counts approach their peak [20]. Our findings also suggest that upper-quartile nevus-counts and clinically-discernible pigmentary characteristics (including fair-skin, freckling, solar lentigines, nevi ≥ 3 mm) evident by age 7 years in Caucasian children from tropical Queensland may facilitate early identification of individuals “at-risk” of melanoma, particularly given that definitive genotyping is not yet routinely available, and may be of limited use in determining non-familial melanoma-risk [25].

Our findings are consistent with those of a shorter-term cohort study conducted in Framingham (latitude 42.47° N), U.S.A. that showed that nevus-count and the presence of large nevi by age 14 years was related to nevus-proneness in late adolescence [26]. Our findings advance current knowledge by demonstrating that nevus-proneness and characteristics indicative of elevated melanoma-risk are evident before age 7 years, at least in Caucasians from tropical Australia’s UV-intense environment. Identifying those “at-risk” early in life provides greater opportunity to intervene with personalized sun-protection counselling in time to reduce risk [26,27] by slowing the rate of nevus-acquisition, which evidence suggests is a modifiable risk-factor [12,13]. 

Although twin studies suggest that up to 68% of the variation in nevus-density can be explained by genetic factors [28,29], 25% of the variation is due to sun-exposure [30]. Childhood is a period of susceptibility to the long-term harmful effects of sun-exposure [7], as evidenced by differences in melanoma-risk between child and adult migrants moving from high to lower latitudes [3,7]. Our results support the hypothesis that early sunburn influences the proliferation of melanocytic nevi, increasing melanoma-risk in sun-sensitive individuals [3,4], and are consistent with earlier findings from our entire cohort [5,6,8]; those of the SONIC childhood nevus cohort in the USA [7,26]; and the transient dermoscopic changes observed in UV-irradiated nevi [31], as well as a pooled-analysis of 5700 cases and 7216 controls (drawn from 13 studies) demonstrating that childhood sunburn increases melanoma-risk for the trunk and limbs by 50% and the head and neck by 40% [32]. However, lifetime number of painful sunburns experienced in the first three decades was not significantly related to the number of incident nevi of all sizes (*p* = 0.079) or ≥2 mm (*p* = 0.329) in our study, even though the trend was indicative of a dose-response. This finding contrasts a meta-analysis that found that melanoma risk continued to increase with number of sunburns across the life-span [33]. Consequently, it is a concern that such high proportions of children (46%) and adults from Queensland (54%) are sunburnt each year [34].

In addition to providing several readily-discernible clues (e.g., presence of freckling, some nevi ≥ 3 mm; numerous nevi, etc.) to aid the early identification of children at increased future risk of melanoma, our findings should inform targeted-screening for melanoma, which is widely accepted as more feasible and cost-effective than population screening [35,36]. This is particularly important in Australia where approximately 1 in 17 (5.9%) people will develop a melanoma before the age of 85 [37]. 

Our findings have the potential to help triage melanoma-prone populations into low and high-risk groups, so that the latter can be targeted with primary prevention and skin surveillance programs (ideally funded by the health care savings they generate) [35,36,37]. Existing training protocols [19,38,39] could be adapted to enable family physicians, pediatricians, physician’s assistants, nurse practitioners, immunizers, community child-health, and/or school nurses working in melanoma-prone populations who routinely see young children (and perhaps parents [38]) to recognize children who display these clinical features. Children identified as “at-risk”, and their parents [38], should be provided with tailored sun-protection counselling [27] and taught the “Ugly Duckling Sign” [39] and skin self-examination [40], promoting these as life-long habits and prompting nevus-prone individuals to self-present for skin-checks [26,27,39,40]. A recent melanoma screening summit in Queensland highlighted the need to compare the “benefits, harms and costs” of changing from opportunistic screening to a more systematic approach for diagnosing melanoma in the face of rising costs and emerging technologies [41]. The most efficacious and cost-effective age to commence targeted risk-based screening, and comparison of the various means by which screening could be delivered needs to considered given the range of possibilities available, from family physicians performing skin-checks using a simplified dermoscopy algorithm [42], to utilizing the growing network of digital imaging systems, and the evolving influence of artificial intelligence in dermatology [41,43]. The advent of electronic medical records in melanoma-prone populations, such as Australia [44], may also help maintain contact with “at-risk” individuals over time.

Parents of our participants completed questionnaires about their child’s sun-exposure history and sun-protection habits annually (on at least two occasions in the 1990s), at a time when most children were under close parental supervision, as few of them attended full-time childcare. Consequently, our estimates of early childhood sun-exposure are less likely to be prone to recall bias than retrospective self-reported estimates of early childhood sun-exposure in case-control studies. Case-control studies of melanoma also collect self-reports of early childhood sun-exposure many decades later, usually without input from participants’ parents, who may no longer be alive. Recall bias is a systematic error that can also occur if the accuracy of a participant’s memories are influenced by subsequent experiences (e.g., a diagnosis of melanoma). Selection bias is also unlikely to explain our results given that our follow-up study was conducted in a representative sample of a population-based cohort that was drawn consecutively from maternity records that were verified using the electoral roll, and had a high participation rate of 87.6%. Consequently, our study is far more representative of the general population from which it was drawn than the only other study to enumerate nevi in childhood and again 20 years later [14]. All participants re-examined for nevi after 20 years in the study by Vrendenborg and co-workers were the biological children of patients diagnosed with familial melanoma (recruited by chance because they accompanied their parents to the clinic) [14]. As immediate family of this hospital-based cohort, the convenience sample of participants recruited by Vrendenborg et al. [14] was far more likely to develop melanoma than the general population of the Netherlands [14].

The main limitation of our 20+ year follow-up is the relatively small sample size due to the time restrictions imposed for this honors research project and limited funding. It has, however, yielded results of value based on a sample of 51 Caucasian participants that closely represents the original cohort [6], while demonstrating the feasibility of continuing to follow this cohort. Although social media is commonly used to recruit research participants [45], the response to unsolicited Facebook messages in this follow-up study was poor because of authenticity concerns. Greater success was achieved by telephoning participants or their parents (particularly after-hours), and using multiple search methods, although laborious.

A larger sample size would have provided greater statistical power and enabled us to control for potential confounders in multivariable analysis. Lack of statistical power may also explain why some explanatory variables were statistically significant for total nevi, but not when counts were limited to nevi ≥ 2 mm. We hope to continue this study and expand the follow-up to include the entire cohort (ideally using 3D sequential digital dermoscopy imaging equipment) in the future (funds permitting), thus enabling us to overcome these limitations.

Further research is required to confirm that the clinical characteristics identified in this study (±family history of melanoma) can discriminate between children at high and low future-risk of melanoma. Children have fewer nevi than adults, making screening for risk-determination quicker and presumably more cost-effective. This is especially relevant in high risk populations such as regional Australia, where access to specialist dermatology care is limited, and general practitioners diagnose the majority of melanomas [37]. Validating these findings in other melanoma-prone populations would provide assurance about the generalizability of our results, including the ideal age to enumerate nevi in Caucasian children for risk-determination purposes. If confirmed, our findings, together with popularization of digital dermoscopy imaging systems [41,43], should inform tailored prevention strategies and yield more cost-effective and efficacious targeted-screening for melanoma.

## 5. Conclusions

We found that young Caucasian Australian children with high nevus-counts tend to become nevus-prone adults, with the occurrence of at least one sunburn before school-age appearing to accelerate the rate of acquisition of new nevi. In addition to having numerous nevi compared to other Caucasian children of the same age, these individuals tend to exhibit clinical features (including freckles, fair-skin, solar lentigines across the shoulders and some nevi ≥ 3 mm in diameter), that distinguish them from others as being at higher lifetime risk of melanoma, around the time they reach school-age. High mole-counts and clinically-discernible features evident in the first decade of childhood may identify at-risk individuals early enough for tailored sun-protection counselling to reduce their risk. Furthermore, if future research confirms that these attributes reliably separate young Caucasian children into risk groups, this should inform more efficacious and cost-effective primary prevention and targeted screening strategies for melanoma, particularly in those populations where the incidence of melanoma is high.

## Figures and Tables

**Table 1 ijerph-17-08680-t001:** Distribution of total melanocytic nevus counts by age in months (m) at baseline for *n* = 51 participants followed-up in 2016.

	1 Year-Old (12–23 m	2 Years-Old (24–35 m)	3 Years-Old (36–47 m)	4 Years-Old (48–59 m)	5 Years-Old (60–71 m)	6 Years-Old (72–83 m)	Total
Mean ± SD	6.9 ± 5.3	14.5 ± 7.4	34.4 ± 19.6	72.9 ± 47.2	70.6 ± 33.3	58.3 ± 23.1	38.3 ± 34.7
Median [IQR]	5.5 [2.25, 10.5]	16 [8.5, 20.5]	36 [16, 50.25]	62 [43.25, 83.25]	60 [43.5, 103]	60 [38.5, 75.5]	32 [13, 55]

**Table 2 ijerph-17-08680-t002:** Demographic and phenotypic characteristics of participants from the Townsville cohort who were available for follow-up and were re-examined for melanocytic nevi (MN), January–June 2016.

Characteristic	*n* (%)	Median Incident MN All Sizes [IQR]	*p*-Value	Median Incident MN 2+ mm [IQR]	*p*-Value
Age at follow-up (years)					
21	5 (9.8)	148 [76, 239.5]		100 [40.5, 166]	
25	5 (9.8)	214 [183, 279]		47 [36.5, 115.5]	
26	6 (11.8)	111 [54.25, 240]		27.5 [7, 42.25]	
27	10 (19.6)	172 [105.25, 268]		43 [27.5, 71]	
28	9 (17.6)	424 [188, 477]		88 [55.5, 186]	
29	6 (11.8)	164 [95.75, 538.25]	(KW)	77.5 [46.8, 186.25]	(KW)
≥30	10 (19.6)	159 [85.75, 351.5]	0.558	39.5 [25, 44.75]	0.012
Gender					
Female	28 (54.9)	193 [110.25, 258.75]	(MW)	54.5 [28.25, 113.75]	(MW)
Male	23 (45.1)	181 [106, 321]	0.955	47 [33, 80]	0.719
Eye color (as an adult) ^3^					
Brown	17 (33.3)	170 [94.5, 276]		40 [26.5, 102]	
Hazel	11 (21.6)	151 [106, 259]	(KW)	40 [28, 74]	(KW)
Blue/green/grey	23 (45.1)	221 [122, 424]	0.491	57 [33, 127]	0.347
Hair color at 18 years					
Black/Dark Brown	24 (47)	147 [77.25, 257.5]		33 [23.5, 77.5]	
Light Brown	19 (37.3)	221 [132, 342]	(KW)	73 [41, 125]	(KW)
Blond	8 (15.7)	235.5 [120.25, 382.5]	0.139	55.5 [34.5, 175.75]	0.021
Natural skin color					
Olive	4 (7.8)	53.5 [43, 211]		7.5 [4.75, 29.75]	
Medium	10 (19.6)	129 [70.5, 214.5]	(KW)	39.5 [24.5, 62]	(KW)
Fair	37 (72.5)	233 [135, 383]	0.013 ^1^	59 [135, 383]	0.003 ^1^
Tendency to burn (as an adult) ^3^					
Tan only	2 (3.9)	62.5 [-, -]		12 [-, -]	
Burn then tan	30 (58.8)	168.5 [105.25, 297]	(KW)	46.5 [31.75, 92]	(KW)
Always burn never tan	19 (37.3)	233 [120, 331]	0.082	57 [30, 125]	0.076
Tanning ability (as an adult) ^3^					
Very brown and deeply tanned	6 (11.8)	86.5 [55.75, 122.75]		25.5 [13.75, 39.25]	
Moderately tanned	22 (43.1)	162 [113, 351.5]		46.5 [33.75, 87.5]	
Slightly tanned	14 (27.5)	245.5 [135.25, 295]	(KW)	71.5 [32.25, 109.25]	(KW)
Not suntanned at all	9 (17.6)	233 [152.5, 383]	0.036 ^1^	57 [38, 157]	0.032
Tanning ability (as a child) ^2,4^					
Very brown and deeply tanned	5 (10.0)	157 [68, 229.5]		39 [22, 86.5]	
Moderately tanned	31 (62.0)	167 [100, 259]		40 [27, 100]	
Slightly tanned	7 (14)	181 [106, 439]	(KW)	74 [40, 88]	(KW)
Not suntanned at all	7 (14)	331 [258, 424]	0.063	52 [47, 207]	0.321
Freckling (as an adult) ^3^					
Absent	16 (31.4)	99.5 [61.5, 141.75]	(MW)	39 [17.75, 72.25]	(MW)
Present	35 (68.6)	255 [167, 424]	0.0001 ^1^	56 [33, 125]	0.024
Freckling (as a child) ^4^					
Absent	23 (45.1)	138 [73, 216]	(MW)	40 [23, 74]	(MW)
Present	28 (54.9)	258.5 [152.75, 414.75]	0.002 ^1^	54.5 [37.5, 113.75]	0.123
Extent of freckling (as a child) ^4^					
Absent	23 (45.1)	138 [73, 216]		40 [23, 74]	
Freckling score 1–20	19 (37.25)	258 [120, 331]		57 [37, 117]	
Freckling score 25–55	6 (11.8)	228 [129, 491.25]	(KW)	46 [26.5, 88.75]	(KW)
Freckling score 60–300	3 (5.9)	443 [-, -]	0.021 ^1^	59 [-, -]	0.315
Score for solar lentigines on shoulders (as an adult) ^3^					
None	23 (45.1)	120 [63, 185]		40 [23, 74]	
10	12 (23.5)	168.5 [121.75, 270.25]		48.5 [29.25, 103]	
20	8 (15.7)	255.5 [234.25, 383.75]	(KW)	44.4 [29.25, 80.75]	(KW)
30–80	8 (15.7)	336.5 [265, 458.75]	0.001 ^1^	78.5 [47.5, 173.5]	0.317
Solar lentigines present on shoulders (as an adult) **^3^**					
Absent	23 (45.1)	120 [63, 185]	(MW)	40 [23, 74]	(MW)
Present	28 (54.9)	256.5 [172.75, 339.25]	0.0001 ^1^	52.5 [30.75, 104]	0.264
Solar lentigines present on shoulders (as a child) ^4^					
Absent	49 (96.1)	181 [106.5, 273]	(MW)	52 [28.5, 94]	(MW)
Present	2 (3.9)	504 [-, -]	0.031 ^1^	89.5 [-, -]	0.536
MN 5+ mm (as a child) ^4^					
Absent	44 (86.3)	183 [106.25, 278]	(MW)	52 [30, 97]	(MW)
Present	7 (13.7)	253 [148, 443]	0.397	41 [28, 139]	1.000
MN 3+ mm (as a child) ^4^					
Absent	24 (47.1)	183 [89.5, 249.5]	(MW)	37.5 [27.25, 73]	(MW)
Present	27 (52.9)	221 [112, 443]	0.025 ^1^	57 [40, 125]	0.083
MN on back (as a child) ^4^					
Absent	19 (37.3)	148 [100, 221]	(MW)	40 [27, 74]	(MW)
Present	32 (62.7)	245.5 [108.25, 435.25]	0.069	54.5 [31.75, 103]	0.242
First degree relative with 50+ MN ^3^					
No	39 (76.5)	170 [100, 263]	(MW)	47 [29, 82]	(MW)
Yes	12 (23.5)	235.5 [159.5, 457.75]	0.069	66.5 [30.25, 123]	0.362
Family history of melanoma ^3^					
Absent	37 (72.5)	157 [106.5, 256.5]	(MW)	41 [28.5, 85]	(MW)
Present	14 (27.5)	312.5 [152.25, 491.25]	0.044 ^1^	56 [37, 185.5]	0.246

IQR Inter-Quartile Range; ^1^ statistically significant after adjustment for multiple testing. KW Kruskal-Wallis Test; ^2^ missing questionnaire data for this variable for *n* = 1 participant. MW Mann-Whitney *U*-Test; ^3^ characteristic determined during follow-up skin examination or questionnaire completed during adulthood. [-, -] Sample size too small to compute IQR; ^4^ characteristic determined at baseline when the participant was a child.

**Table 3 ijerph-17-08680-t003:** Annual rate of acquisition of new melanocytic nevi (MN) acquired since baseline shown by total MN prevalence (expressed as quartiles) at baseline.

Quartile of MN within Age Group at Baseline	*n* (%)	Total MN Count at Baseline (Prevalence of MN as a Child)			Annual MN Incidence Rate (New MN/Year)			*p*-Value
		Mean ± SD	Median	[IQR]	Mean ± SD	Median	[IQR]	
0–25th percentile (Bottom Quartile)	12 (23.5)	17.3 ± 4.4	17.5	[3, 30.25]	5.7 ± 3.7	4.7	[2.5, 8.7]	
25–49th percentile	12 (23.5)	34.3 ± 23.3	38.5	[13.25, 54]	6.6 ± 7.1	6.6	[4.7, 10.5]	Rho = 0.499
50–74th percentile	15 (29.5)	40.5 ± 26.9	42	[16, 55]	11.0 ± 4.9	10.4	[8.2, 14.3]	*p* < 0.0005 ^1^
75–100th percentile (Upper Quartile)	12 (23.5)	60.6 ± 52.4	63	[13.5, 84]	13.2 ± 8.8	13.3	[4.4, 20.4]	

Rho represents Spearman’s rank correlation coefficient. ^1^ statistically significant, *p* ≤ 0.05.

**Table 4 ijerph-17-08680-t004:** Median incident melanocytic nevi (MN) and median incident MN ≥ 2 mm by sun-exposure variables.

	*n* (%)	Median Incident MN All Sizes [IQR]	*p*-Value	Median Incident MN 2+ mm [IQR]	*p*-Value
Sunburn age < 7 years					
No	20 (39.2)	126 [64, 205.8]		38 [17.8, 73]	(MW)
Yes	31 (60.8)	238 [151, 424]	(MW) 0.003 ^1^	56 [39, 125]	0.022
Frequency of erythema age < 7 years					
Nil	21 (41.2)	132 [67, 233.8]		39 [19.5, 72]	
Once or twice	22 (43.4)	218.5 [134, 265]		52.5 [28.8, 125.5]	(KW)
Annually (1–4 times/yr)	8 (15.7)	380 [190, 478.3]	(KW) 0.008 ^1^	94 [44.8, 133.5]	0.034
Peeling/blistering sunburn age < 7 years					
No	36 (70.6)	157.5 [92.3, 254.5]		43.5 [27.25, 95.5]	(KW)
Yes	15 (29.4)	321 [157, 490]	(MW) 0.003 ^1^	53 [40, 117]	0.260
Average time spent outdoors during primary school years					
<1 h/day	3 (5.9)	107 [-, -]		40 [-, -]	
1–2 h/day	5 (9.8)	181 [112.5, 312.5]		29 [16, 42.5]	(KW)
>2–6 h/day	43 (84.3)	216 [112, 289]	(KW) 0.774	56 [36, 104]	0.125
Average time spent outdoors during high school years					
<1 h/day	6 (11.8)	338.5 [160.8, 470.5]		41 [23.75, 185.5]	
1–2 h/day	12 (23.5)	163.5 [108.25, 221]		58 [29.25, 112.75]	(KW)
>2–6 h/day	33 (64.7)	214 [103, 326]	(KW) 0.252	49 [32, 85]	0.952
Main leisure activities since school					
Mainly indoors	12 (23.5)	129 [79.5, 184]		39.5 [30, 106]	
Both indoors and outdoors	32 (62.8)	224.5 [114.5, 339.3]		50.5 [28, 97]	(KW)
Mainly outdoors	7 (13.7)	238 [132, 289]	(KW) 0.109	59 [37, 125]	0.750
Main occupations since school					
Mainly indoors	28 (54.9)	193 [114, 277]		71.5 [29.25, 113.75]	
Both indoors and outdoors	18 (35.5)	193 [97.5, 323.5]		44 [27.75, 53.75]	(KW)
Mainly outdoors	5 (9.8)	181 [96.5, 353]	(KW) 0.977	37 [28, 106.5]	0.347
Time spent outdoors on typical weekdays in the previous year					
Hardly ever (≤1 h/day)	24 (47.1)	140.5 [99.25, 225]		48.5 [27.25, 86.5]	
≤50% of time (2–4 /day)	18 (35.3)	256.5 [158, 456]		56 [39.75, 156]	(KW)
>50% of time (5+ h/day)	9 (17.7)	216 [149.5, 305]	(KW) 0.032 ^1^	40 [30, 51.5]	0.127
Time spent outdoors on typical weekends in previous year					
Hardly ever (≤1 h/day)	9 (17.65)	181 [109.5, 345.5]		57 [31, 157]	
≤50% of time (2–4 h/day)	33 (64.7)	167 [92.5, 261]		40 [27, 85]	(KW)
>50% of time (5+ h/day)	9 (17.65)	283 [168, 383]	(KW) 0.272	56 [49.5, 92]	0.254
Lifetime history of painful sunburn reported as an adult					
Never	0 (0)	-		0 (0)	
Once	3 (5.9)	61 [-, -]		23 [-, -]	
2–5 times	28 (54.9)	183 [110.25, 254.5]		43.5 [27.5, 104]	
5–10 times	13 (25.5)	151 [105.25, 268]		52 [39, 77]	(KW)
>10 times	7 (13.7)	424 [170, 490]	(KW) 0.079	56 [40, 139]	0.329

IQR Inter-Quartile Range; [-, -] Sample size too small to compute IQR; KW Kruskal–Wallis Test; ^1^ statistically significant after adjustment for multiple testing MW Mann–Whitney *U*-Test.

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
