# Peer review of "Clinical Characteristics in Early Childhood Associated with a Nevus-Prone Phenotype in Adults from Tropical Australia: Two Decades of Follow-Up of the Townsville Preschool Cohort Study"

_ijerph, 2020, doi:10.3390/ijerph17228680_

Round 1

Reviewer 1 Report

The authors have correctly adjusted and commented on each of the points raised. The tables are easier to understand and an appropriate size.

In general, the manuscript has been made clearer, although there are some sections that are still too long, such as the statistics and some of the corrections made that could be explained more succinctly.

Author Response

Dear Editorial team,

Thank you for allowing us the time required to address this list of minor changes satisfactorily.

We have reduced the length of the statistics section considerably (whilst attending to the changes required in relation to the academic editor’s questions). We have also tried to shorten our responses in the text to make these more succinct where possible, without reducing clarity.

All changes aimed at reducing the word count have been marked using track changes in the most recently revised version of the manuscript and can easily be identified as they are red while my corrections from the previous round of comments are marked up in track changes in blue.

Kind Regards,

Simone

This manuscript is a resubmission of an earlier submission. The following is a list of the peer review reports and author responses from that submission.

Round 1

Reviewer 1 Report

The study by Barsoum and Harrison compares nevus counts in children with nevus counts from a re-examination 20 years later in the same cohort. The aim was to investigate the association between the two nevi counts and identify possible predictors for nevi-prone adults, facilitating more targeted screening and information of high-risk groups for melanoma early in life.

Overall, the manuscript addresses an interesting aspect of melanoma prevention and describes the study appropriately. There are still some points especially concerning the statistical methods and presentation of the results that should be clarified or addressed.

Main points:

  • Please address the problem of false positive test results due to multiple testing and describe how this problem was dealt with in the analyses.
  • P-values between 0.05 and 0.1 should not be interpreted as „borderline significant“. If the significance level is set to 0.05, a p-value >0.05 is not significant.

Minor points:

  • Abstract: it is not always clear that the numbers in the abstract refer to the incident/aquired nevi, e.g. line 21/22 „sunburn before 7 years old resulted in higher adult nevus counts (238 vs 126)“ -> that could be misinterpreted as prevalent nevus counts in adults.
  • Line 20: the p-value here is <0.0005, but in Table 2 is p=0.037. Which one is correct?
  • Line 107/108: what were the elements of the review of nevi ≥5mm by Prof Soyer? Why were only photographs of those nevi reviewed?
  • Line 152/153: „median age was 3.2 [IQR 2.3, 2.9]“ -> please check the numbers, the median can not be higher than the 75% quartile.
  • Line 154: „majority of participants had light eyes“ -> according to Table 1 only n=12 had light eyes (but since 12 of 51 is also not 45.1%, the number there is probably wrong)
  • I’d recommend checking the numbers in the tables carefully, since there were several typos, e.g. Table 1 Current employment / Home duties: 9 of 51 = 17.6%; solar lentigines on shoulders: 49 of 51 = 96.1%; Table 3 50-74th percentile: 15 of 51 = 29.4% etc.
  • If the sample size was to small to compute the IQR, it is confusing to give only half of it ("[8, - ]"), better report completely as [-,-]
  • The age of the participants at the re-examination is given in the abstract as 21-30, however in Table 1 and 2 it is „>=30“. What was the maximum age?
  • Table 1: „Tanning ability (adult)“: this characteristic was determined at baseline, does „(adult)“ therefore refer to the parent?
  • Table 1: „Solar lentigines on shoulders (child)“: please specify which test is meant by „Exact“
  • Table 2: „25 years“ results for MN ≥5mm -> IQR is given with „[0.6.5]“, there is a „,“ missing
  • Table 2: How was the IQR calculated? Because if the number of prevalent MN was measured on a discrete scale, the 25% / 75% percentile could not be „.3“ or „.8“
  • Line 167/168: „54.9% employed in predominantly outdoor occupations“: according to Table 4, 54.9% were in predominantly indoor occupations
  • Line 177: „poor tanning ability“ -> which variable is meant by that? „Ability to tan (child)“ or „tanning ability (child)“ are not significantly associated with higher counts of new nevi according to table 1
  • Line 203 -208: paragraph is repeated (from line 195-202)

Reviewer 2 Report

  1. The part of the discussion in relation to the digital follow up strategies and sun counselling it is too long.

  1. Is Doctor RB a dermatologist? It is not clear to me…If he’s not, how was the training for him? Did he use dermoscopy to evaluate the lesions? Which criteria did he use to differentiate nevus less than 2mm and freckles?

  1. Why did the authors differentiate between nevi less and more than 2mm? It should be explained.

  1. I would like to see in any of the table the number of nevi by age group in the baseline. Maybe it would make it easier to understand some results and to evaluate If the prevalence is comparable to other studies conducted in Australian population.

  1. The prevalence of melanoma in the study in relation to general population in Australia is higher. I think that the explanation could be the small sample size but it could be also a bias in the selection. The percentage of the patients with melanoma family history or 50+ nevialso seems high. The author says this is the difference with the other study published, so I think this should be better explained.

  1. They talk about painful sunburns in adulthood. Why painful sunburns? Why weren’t sunburns assessed as in childhood (erythema, peeling/blistering)?

  1. What  is the  difference between ability to tan and tanning ability in Table 1? They mention in the results an almost significant  relation  between poor ability to tan and higher incidence of nevi but it is  confusing  in the table.

  1. In Table 1,  in part “age at follow-up”, significative p-value corresponds to which comparison? Between all age groups? Between 21 and 30? It should be explained .

  1. In Table 1, there are some results that are irrelevant for the main objective of the study as highest maternal education, highest qualification, current employment etc.I can imagine this data could be important in order to understand the sun exposure behaviour, but the authors don’t comment  on this, so I think it is not necessary to put in the table  as it is too long.

  1. The part about the recrution of patients can be shortened.

  1. Table 2 is too long and gives little information.

To be freckle it is   significant in children in the total group but not in the <2mm group. How do they explain this result? Also in the case of solar lentigos…

Maybe, freckled children have more incident freckles and  these are counted as nevi <2mm? It should be  clarified.

  1. There are many results where the result is significant for one group of nevi and not for the other…How  are these differences explained??

I think it is an interesting study because as the authors says, there are no  other studies with a follow-up in a cohort after 20 years and also because they have  significant results as the relation with sunburns that generates some controversies nowadays, but the results should be better explain and commented at discussion. The is a lot of data in the tables but short comments about it. These could be reviewed.
